# Pretreatment with Group II Metabotropic Glutamate Receptor Agonist LY379268 Protects Neonatal Rat Brains from Oxidative Stress in an Experimental Model of Birth Asphyxia

**DOI:** 10.3390/brainsci8030048

**Published:** 2018-03-17

**Authors:** Ewelina Bratek, Apolonia Ziembowicz, Elzbieta Salinska

**Affiliations:** Department of Neurochemistry, Mossakowski Medical Research Centre, Polish Academy of Sciences, 02-106 Warsaw, Poland; ebratek@imdik.pan.pl (E.B.); aziembowicz@imdik.pan.pl (A.Z.)

**Keywords:** hypoxia-ischemia, birth asphyxia, group II metabotropic glutamate receptors, LY379268, neuroprotection, oxidative stress

## Abstract

Hypoxia-ischemia (H-I) at the time of birth may cause neonatal death or lead to persistent brain damage. The search for an effective treatment of asphyxiated infants has not resulted in an effective therapy, and hypothermia remains the only available therapeutic strategy. Among possible experimental therapies, the induction of ischemic tolerance is promising. Recent investigations have shown that activation of group II metabotropic glutamate receptors (mGluR2/3) can provide neuroprotection against H-I, but the mechanism of this effect is not clear. The aim of this study was to investigate whether an mGluR2/3 agonist applied before H-I reduces brain damage in an experimental model of birth asphyxia and whether a decrease in oxidative stress plays a role in neuroprotection. Neonatal H-I on seven-day-old rats was used as an experimental model of birth asphyxia. Rats were injected intraperitoneally with the mGluR2/3 agonist LY379268 24 or 1 h before H-I (5 mg/kg). LY379268 reduced the infarct area in the ischemic hemisphere. Application of the agonist at both times also reduced the elevated levels of reactive oxygen species (ROS) in the ipsilateral hemisphere observed after H-I and prevented the increase in antioxidant enzyme activity in the injured hemisphere. The decrease in glutathione (GSH) level was also restored after agonist application. The results suggest that the neuroprotective mechanisms triggered by the activation of mGluR2/3 before H-I act through the decrease of glutamate release and its extracellular concentration resulting in the inhibition of ROS production and reduction of oxidative stress. This, rather than induction of ischemic tolerance, is probably the main mechanism involved in the observed neuroprotection.

## 1. Introduction

Birth asphyxia or hypoxia-ischemia (H-I) at birth resulting in brain damage is still an important problem in modern obstetrics, even in developed countries. The encephalopathy that follows a hypoxic-ischemic insult reflects an evolving process characterized by an initial primary injury followed by a self-sustaining cascade of harmful biochemical events that lead to further brain damage.

The mechanism of H-I brain injury involves the increased release and extracellular retention of glutamate and excessive activation of ionotropic glutamate receptors, especially the *N*-methyl-d-aspartate (NMDA) receptor, leading to excitotoxicity. The insufficient oxygen and glucose supply during H-I also results in the accumulation of toxic products such as reactive oxygen species (ROS). The accumulation of ROS initiates oxidative stress, shifting the balance between oxidants and antioxidants in favour of oxidants. This shift leads to the oxidation of cell lipids, proteins, and other constituents, resulting in cell injury and death [1,2]. Cells are equipped with enzymes that, in physiological conditions, are sufficient to combat oxidative stress and to neutralize ROS. However, in hypoxic-ischemic conditions, even increased activity of superoxide dismutase (SOD), catalase (CAT), and glutathione peroxidase (GPx) supported by glutathione cannot eliminate the overloading concentrations of ROS [3].

The search for an effective treatment of asphyxiated infants has been conducted by many scientists for years. However, despite many experimental methods of neuroprotection proposed by scientists, hypothermia remains the only available therapeutic strategy as a treatment of asphyxiated infants. Most of the drugs appear to have short-term benefits in the treatment of anoxic brain injury and no specific drug therapy has been shown to improve long-term survival in controlled trials [4]. Therefore, there is a need to develop additional strategies that may provide neuroprotection and be combined with other therapies.

One of the methods considered recently as a potential therapy against hypoxic-ischemic brain damage is the induction of endogenous protective mechanisms in the brain. These mechanisms may be triggered by preconditioning with a sublethal stress that induces resistance to subsequent lethal stress. Preconditioning with ischemia, hypoxia, mild oxidative stress, or oxidative phosphorylation inhibition can activate ischemic tolerance when applied below the damage threshold [5,6,7]. However, as perinatal H-I is unpredictable and such manipulations may be potentially dangerous, they seem not to have a credible translational potential in clinical practice [4]. Consequently, safe stimuli with the potential to trigger neuroprotective mechanisms are still a subject of extensive investigation.

Group II metabotropic glutamate receptors and their role of in H-I brain injury have gained increasing attention. Metabotropic glutamate receptor 2 (mGluR2) and 3 (mGluR3), negatively coupled to cyclic AMP formation, act as presynaptic autoreceptors that regulate glutamate transmission and therefore appear to be promising targets for the induction of neuroprotection. Because of the role of group II mGluRs in the monitoring of any excessive glutamate that escapes from the synaptic active zone, their activation and suppression of glutamate release may have an important impact on the development of H-I injury [8,9].

The neuroprotective effects of selective agonists of mGluR2/3 against ischemic brain injury have been shown in a number of animal studies. The neuroprotective effects of a mGluR2/3 agonist (-)-2-oxa-4-aminobicyclo[3.1.0]hexane-4,6-dicarboxylic acid (LY379268), applied in a short time after the insult, were reported in an experimental model of birth asphyxia in rats and the global ischemia in gerbils [10,11]. Recently mGluR2/3 has been also shown to be involved in the induction of ischemic tolerance [12,13]. The reduction in glutamate release appears to be an obvious element of the neuroprotection mediated by mGluR2/3 activation; however, the exact mechanism of this neuroprotection is still not fully understood and seems to be complex.

The relationship between mGluRs and oxidative stress was demonstrated many years ago; however, there are relatively few studies in the literature investigating the role of mGluR2/3 in antioxidant defence. The activation of group II mGluRs has been shown to attenuate oxidative stress-induced cell death in spinal cord injury [14], protect neurons from glucose-induced oxidative injury [15], and reduce ROS production in the immature rat brain during seizures induced by a bilateral intracerebroventricular infusion of dl-homocysteic acid [16]. Recently, activation of group II mGluRs was also shown to protect against the ischemia-induced free radical programmed death of rat brain endothelial cells, although this neuroprotective effect was considered to be a complex metabotropic glutamate receptor response [17]. The exact mechanism by which mGluR2/3 interacts with ROS production and inactivation is not clear. Berent-Spillson and Russell [15] have shown that activation of mGluR3 increases the concentration of glutathione, an important element of antioxidant cell defence. However, there is no information concerning the effect of group II mGluRs activation on changes in antioxidant enzyme activity resulting from ischemic insults.

Most of the data mentioned above were obtained from experiments where mGluR2/3 agonists were applied after ischemic insult. To the best of our knowledge there is no data concerning the effect of mGluR2/3 application before H-I on immature animals or the possible induction of brain ischemic tolerance.

Therefore, assuming that the activation of group II mGluRs before H-I may induce ischemic tolerance and have a neuroprotective effect, we decided to more closely investigate the molecular mechanism(s) of this effect.

The aim of our study was to investigate whether the activation of mGluR2/3 before experimental birth asphyxia on seven-day-old rats will result in the induction of defence mechanisms resulting in neuroprotection and if the antioxidant enzymes are engaged in this process.

## 2. Materials and Methods

All experiments described in this paper were approved by the 4th Local Ethical Committee in Warsaw, Poland and performed in accordance with Polish governmental regulations (Dz.U.97.111.724) and the European Community Council Directive of 24 November 1986 (86/609/EEC). All efforts were made to minimize animal suffering and the number of animals used.

### 2.1. Induction of Cerebral Hypoxia–Ischemia

Neonatal cerebral H-I was induced according to Rice et al. (1981). Briefly, seven-day-old Wistar rat pups (weight 12–18 g) were anaesthetized with isoflurane (4% for induction, and 1.5–2.0% for maintenance) in a mixture of nitrous oxide and oxygen (0.6:1). The left common carotid artery was exposed and cut between double ligatures of silk sutures or only exposed (sham control). Animals were placed in home cages, allowed to recover for 60 min, placed in a humidified chamber (35 °C), and exposed to hypoxic conditions (8% oxygen in nitrogen) for 75 min. This duration of hypoxia-ischemia is typically associated with infarction predominantly of the cerebral hemisphere ipsilateral to the carotid artery occlusion [18,19]. After hypoxic treatment the animals were returned to the cages and housed at room temperature (22 °C) with a 12:12 h light–dark cycle and with ample food and water.

### 2.2. Drug Application

Animals were injected intraperitoneally (i.p.) with LY379268, a highly selective agonist of group II mGluRs, for which its affinity to mGluR2 is two times greater than that to mGluR3. The animals received a single injection made 24 or 1 h before H-I in a dose of 5 mg/kg of body weight. The LY379268 dose was based on previous findings [10,11]. Sham-operated and H-I control rats were injected with saline.

### 2.3. Evaluation of Brain Damage

The chosen animals from each group (*n* = 4–5 per group) were sacrificed at 7 days after the H-I insult for an evaluation of the brain infarct area. Brains were cut into 2 mm thick coronal sections using a vibrating microtome (Campden Instruments Ltd., Loughborough, UK). The sections were stained with 1% 2,3,5-triphenyltetrazolium chloride (TTC, Sigma-Aldrich, St. Louis, MO, USA) for 8 min to measure infarct volume. The slices were then fixed in 4% paraformaldehyde and subjected to further analysis. A computerized video camera-based image analysis system was used to measure the cross-sectional areas of unstained tissue (open source image processing program ImageJ, http://imagej.net/index.html, designed by Wayne Rasband, NIH, USA).

A histochemical evaluation of brain damage was performed on brains isolated 7 days after H-I. Animals were anaesthetized with halothane and subjected to intracranial perfusion fixation with 4% neutralized formalin (Sigma-Aldrich, St. Louis, MO, USA). The brains were then removed, postfixed for 4 h in the same fixative, and embedded in paraffin. Paraffin blocks containing brain tissue were cut in serial sections at 10 µm thickness using a rotator microtome (Hydrax M 25, Zeiss). The sections were stained according to the Nissl protocol with 0.5% cresyl violet for the histological assessment of neuronal cell damage.

### 2.4. Determination of ROS Level

The ROS levels in brain hemispheres were determined using 2,7–dichlorofluorescein diacetate (DCF-DA), a fluorogenic dye which, in contact with ROS, is converted into highly fluorescent 2′,7′–dichlorofluorescin (DCF) detected by fluorescence spectroscopy. The brains were collected 3 h after H-I, and tissues from the left and right hemisphere were homogenized separately in an ice-cold 40 mM Tris-HCL buffer (pH 7.4). The resulting brain homogenates were incubated with DCF-DA (25 μM) for 30 min at 37 °C. The formation of the fluorescent product DCF was monitored by a fluorescence spectrometer with an excitation wavelength of 488 nm and emission wavelength of 525 nm. The relative fluorescence unit (RFU) was calculated per 1 mg protein in homogenate. The effect of H-I and the group II mGluR agonist used was estimated by determination of the ROS level in individual brains and standardized to the basal level measured in brains from sham-operated animals.

### 2.5. Determination of Glutathione Concentration

Determination of glutathione (GSH) concentration was carried out in tissues from left and right hemispheres isolated 3 h after the H-I insult. The left and right hemispheres were homogenized separately in 25 mM HEPES (pH 7.4) containing 250 mM sucrose, then centrifuged at 1000× *g* for 5 min at 4 °C. Supernatants were collected, and the concentration of reduced glutathione was determined in collected supernatants using the Glutathione Assay Kit (Cayman Chemical Company, Ann Arbor, MI, USA) following the manufacturer’s procedure.

### 2.6. Determination of Antioxidant Enzyme Activity

The brains were collected 3 h after the H-I insult, and the left and right hemispheres were homogenized separately in a cold 50 mM potassium phosphate buffer (pH 7.4) containing 1 mM ethylenediaminetetraacetic acid (EDTA) and then centrifuged at 10,000× *g* for 15 min at 4 °C. The catalase activity was determined in the collected supernatant using the Catalase Assay Kit (Cayman Chemical Company, USA) following the manufacturer’s procedure.

For the measurement of SOD activity, the brain tissue was rinsed with PBS buffer (pH 7.4) to remove any blood cells and clots. The left and right hemispheres were homogenized separately in 20 mM HEPES buffer (pH 7.2), containing 1 mM EDTA, 210 mM mannitol, and 70 mM sucrose per gram tissue. Homogenates were centrifuged at 1500× *g* for 5 min at 4 °C. The supernatant was collected and the activity of SOD was determined using the Superoxide Dismutase Assay Kit (Cayman Chemical Company, USA), following the manufacturer-provided instructions.

Glutathione peroxidase (GPx) activity was measured in both brain hemispheres separately. Tissue was homogenized in a cold buffer (50 mM Tris-HCL (pH 7.5), 5 mM EDTA, and 1 mM dithiothreitol (DTT) and centrifuged at 10,000× *g* for 15 min at 4 °C, and the supernatant was collected for the assay. GPx activity was determined using the Glutathione Peroxidase Assay Kit (Cayman Chemical Company, USA).

## 3. Results

### 3.1. LY379268 Reduces H-I Evoked Brain Damage

H-I resulted in brain tissue damage in the ipsilateral brain hemisphere. In the present study the evaluation of the infarct area using TTC staining showed that H-I produced 42% infarction of the ipsilateral hemisphere (Figure 1). LY379268 applied 24 or 1 h before H-I significantly decreased the area of infarction to 18.8% and 8.5%, respectively (*p* < 0.001). However, the infarct area measured after application of LY379268 1 h before H-I was significantly smaller than when the agonist was applied 24 h before the insult (*p* < 0.001).

### 3.2. LY379268 Reduces the H-I Evoked Increase in ROS Level

H-I resulted in an increase in the ROS level in the left ischemic hemisphere by as much as a factor of three (*p* < 0.001; Figure 2). The application of LY379268 at both times significantly reduced the ROS level (*p* < 0.005). LY379268 applied 24 h before H-I decreased the ROS level by 30% (*p* < 0.001), whereas when applied 1 h before H-I, it decreased the ROS level by 45%. However, the difference between the ROS level after agonist application at 24 or 1 h before H-I was not statistically significant. Neither of the treatments changed the ROS level in the right hemisphere.

### 3.3. LY379268 Pretreatment Reduces the H-I Increased Activity of SOD and Catalase

SOD activity in the brain tissue measured 3 h after H-I increased four times compared to the SOD activity in the control sham-operated animals (Figure 3). Application of LY379268 24 h before H-I decreased SOD activity by 25% compared to the H-I group and by 42% when application of the agonist was performed 1 h before H-I.

In response to H-I, catalase activity in the left hemisphere increased almost six times. The application of LY379268 24 h before H-I decreased of catalase activity by 33% compared to the H-I group, whereas application 1 h before H-I decreased catalase activity by 55% (Figure 4). The decrease in SOD and catalase activity caused by LY379268 application 1 h before H-I was significantly different from that measured in the group treated with the agonist 24 h before H-I.

Neither of the treatments changed the SOD or catalase activity in the right hemisphere.

### 3.4. LY379268 Reduces the H-I Evoked Changes in GPx Activity and the GSH Level

The activity of GPx measured in the left hemisphere 3 h after H-I increased 13 times (from 2.9 to 38.3 U/mg protein) compared to that in sham-operated animals (Figure 5a). Application of LY379268 24 h before H-I decreased GPx activity by 23% and by 33% when applied 24 h before H-I. Neither of the treatments changed GPx activity in the right hemisphere.

H-I decreased the GSH concentration in both hemispheres, although the decrease in the left hemisphere was significantly larger than that in the right (to 36% of control in the left and to 60% in the right hemisphere) (Figure 5b).

Application of LY379268 24 or 1 h before H-I reduced the decrease in the GSH concentration in the left hemisphere (to 60% and 67.5%, respectively), although LY379268 did not affect the decrease observed in the right hemisphere.

## 4. Discussion

Agonists of group II mGluRs, such as LY379268, are increasingly being considered as potential neuroprotective drugs. These agonists were recently reported to be potential drugs in the treatment of Alzheimer’s and Parkinson’s diseases, as well as schizophrenia and anxiety [20]. The neuroprotective effect of LY379268 on H-I has mostly been investigated when the application occurred after H-I or global ischemia. The results presented in this paper show that the neuroprotective effect of LY379268 also occurred when the agonist was applied before the H-I. These results are in agreement with the neuroprotection observed by Bond et al. [11] with bilateral common carotid occlusion in the gerbil. Moreover, Bond et al. [11], showed that LY379268 demonstrated prolonged activity and was active even applied 24 and 48 h before ischemia. We observed a similar effect when LY379268 was applied 24 h or 1 h before H-I in immature rats. The beneficial effect of this agonist is associated not only with its action at presynaptic mGluR2 and the prevention of glutamate release but also with activation of mGluR3 on astrocytes and the production and release of transforming growth factors-β1 (TGF-β1) and -β2, which protect cells from apoptosis [21]. Recently, pretreatment with LY379268 was shown to fail to induce expression of specific neurotrophic factors such as TGF-β, BDMF, and NGF [11]. The authors concluded that the neuroprotective effect of LY379268 involved inhibition of glutamate release and inhibition of biochemical pathways leading to programmed cell death. However, the mechanisms of the observed neuroprotective effects were not defined. The experiments presented in this paper examine more closely one of the possible mechanisms of the neuroprotective action of LY379268, its effect on oxidative stress.

Oxidative stress contributes to the pathogenesis of immature brain injury after hypoxia-ischemia. The cascade of toxic events involves glutamate release, activation of NMDA receptors, and calcium ion influx. The activation of NMDA receptors by glutamate is a strong stimulus for release of nitric oxide (NO), a reactive free radical. NO is one of the principal mediators of neuronal degeneration. It was shown that NO, as a free radical, can act at a molecular level that involves DNA degeneration and the apoptosis processes [22].

High concentrations of NO irreversibly inhibit the mitochondrial respiratory chain and diminish the ability of neurons to tolerate oxidative stress. Furthermore, mitochondrial dysfunction leads to superoxide leakage and the formation of other ROS [1,23]. Under these conditions the endogenous antioxidant defence system in both the adult and immature brain responds by increasing the activity of SOD, catalase, and GPx [24,25]. However, the antioxidant defence system in the immature brain is less active; therefore, the identification of a factor that reduces ROS production or increases antioxidant enzymes activity is critical [24].

It was shown that physiological glutamatergic transmission mediated by synaptic NMDA receptors up-regulates the peroxiredoxin-thioredoxin enzyme system, another element of antioxidant defences, whereas increased receptors activity may overwhelm the thioredoxin system [26]. On the other hand, extra-synaptic NMDA receptors activated by the excess glutamate leakage from synaptic cleft results in neurodegenerative mitochondrial superoxide anion generation [27]. Excessive extracellular glutamate can also cause excitotoxicity by the inhibition of cysteine uptake, resulting in a depletion of intracellular GSH and oxidative stress [28].

The presented results show, for the first time, that the neuroprotective effect of pretreatment with the group II mGluRs agonist LY379268 in the immature brain against H-I is strongly connected to inhibition of ROS production. Application of LY379368 24 or 1 h before H-I not only reduced the brain damage in the ipsilateral hemisphere but also suppressed the increase in the ROS level evoked by the insult. The decrease in the antioxidant enzyme activity and partially restored GSH concentration observed in our experiments indicates that the reduction in ROS concentration was the result of not only the efficient neutralization of ROS by these enzymes but also the suppression of ROS production. In addition, the neuroprotective effect was likely mostly caused by LY379268 activation of presynaptic mGluR2 receptors, inhibition of excessive glutamate release, and probably inhibition of neurodegenerative pathways that are dependent upon NO activity and lead to excessive ROS production. These results not only support those reported by Bond et al. [11] on the neuroprotective effect of pretreatment with LY379268 in an experimental model of global ischemia in the gerbil but also show that LY379268 provides neuroprotection in a perinatal H-I model and reveal one of the molecular mechanisms. The exact effect of LY379268 on NO production after H-I and the engagement of thioredoxin in the defence against H-I induced oxidative stress is still to be revealed.

Previously, a preconditioning ischemia, hypoxia, or asphyxia has been shown to trigger defence mechanisms in experimental models of perinatal asphyxia [6,29]. Most of these mechanisms depend on the activation of NMDA receptors. However, as demonstrated by Bond et al. [11] in an experimental model of ischemia in gerbils, LY379268 almost blocked the tolerance induced by preconditioning ischemia and this effect was associated with a reduction in NMDA receptor activation. We agree that pretreatment with LY379268 does not support the induction of ischemic tolerance and mostly acts by decrease in glutamate release and consequent prevention of the molecular cascade leading to oxidative stress. Nevertheless, LY379268 may be a potent factor used to prevent H-I-induced brain damage in the situations of labour and delivery complications where preconditioning methods, still considered to be risky, cannot be used.

The inhibition of glutamate release and the subsequent reduction in H-I-evoked ROS production seems to be the main mechanism of the neuroprotective effect of LY379268 in experimental birth asphyxia. However, these findings do not negate the possibility that activation of mGluR2/3 before H-I may activate other intracellular mechanisms that may inhibit neuronal death.

## 5. Conclusions

The results presented in this paper show that the inhibition of glutamate release and the subsequent reduction in H-I-evoked ROS production seem to be the main mechanism of the neuroprotective effect of LY379268 in experimental birth asphyxia. However, these findings do not negate the possibility that activation of mGluR2/3 before H-I may activate other intracellular mechanisms that may inhibit neuronal death.

## Figures and Tables

**Figure 1 brainsci-08-00048-f001:**
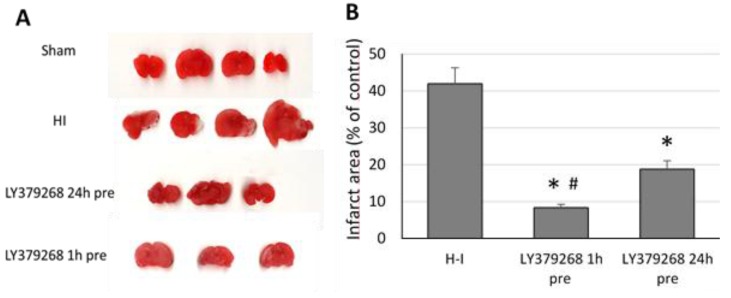
(**a**) Application of LY379268 before hypoxia-ischemia (H-I) reduces the infarct area in the ipsilateral hemisphere. Representative photomicrographs of TTC staining of the infarct areas in the brains of rats; (**b**) Quantification of the infarct area after H-I revealed by TTC staining. Number of animals per group *n* = 4–5. Results are presented as the mean values ± SD, * *p* < 0.001—significantly different from the H-I group; #—significantly different from the group injected 24 h before H-I. The brains were analysed 7 days after H-I.

**Figure 2 brainsci-08-00048-f002:**
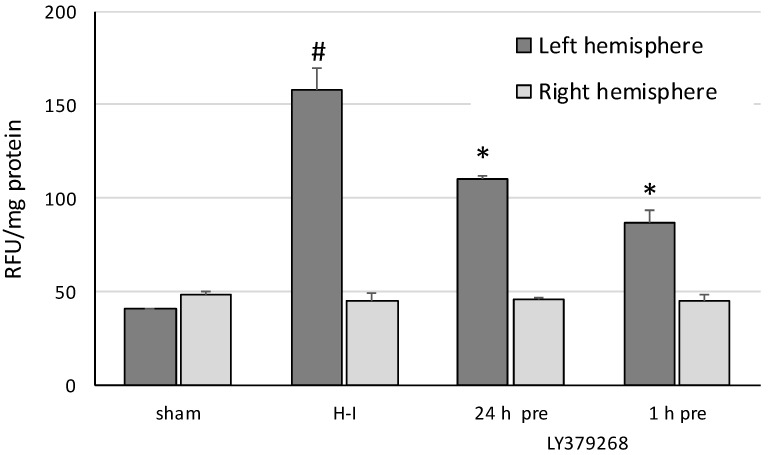
The effect of LY379268 pretreatment on changes in the reactive oxygen species (ROS) level evoked by H-I. The results are presented as the mean ± SD, *n* = 3–6; #—significantly different from the sham-operated, *p* < 0.005; *—significantly different from the H-I group, *p* < 0.005.

**Figure 3 brainsci-08-00048-f003:**
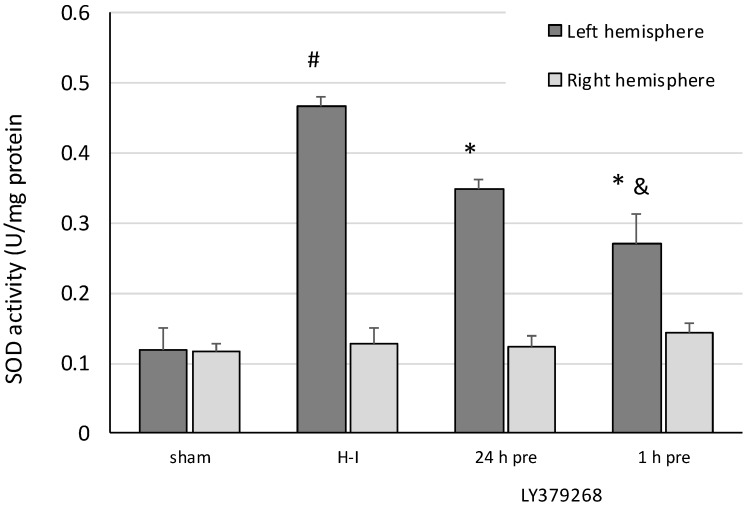
The effect of LY379268 pretreatment on changes in superoxide dismutase (SOD) activity evoked by H-I. The results are presented as the mean ± SD, *n* = 3–5; #—significantly different from the sham-operated group, *p* < 0.005; *—different from the H-I group, *p* < 0.005; &—different from the group injected 24 h before H-I, *p <* 0.01.

**Figure 4 brainsci-08-00048-f004:**
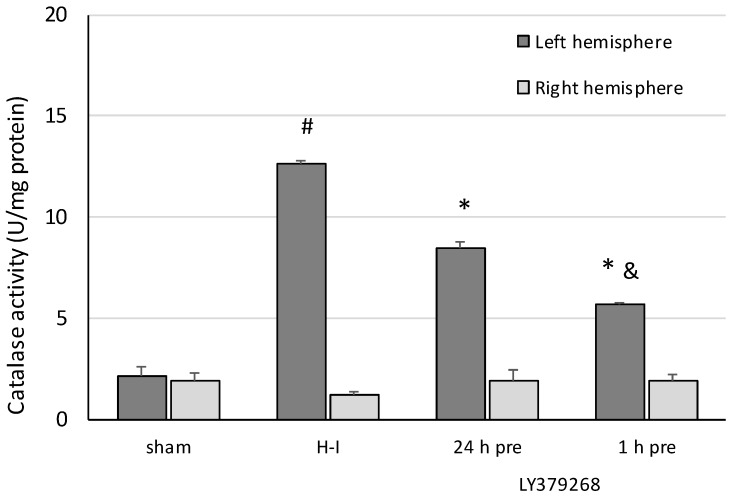
The effect of LY379268 pretreatment on changes in catalase activity evoked by H-I. The results are presented as the mean ± SD, *n* = 3–5; #—significantly different from the sham-operated group, *p* < 0.005; *—different from the H-I group, *p* < 0.005; &—different from the group injected 24 h before H-I, *p < 0.01*.

**Figure 5 brainsci-08-00048-f005:**
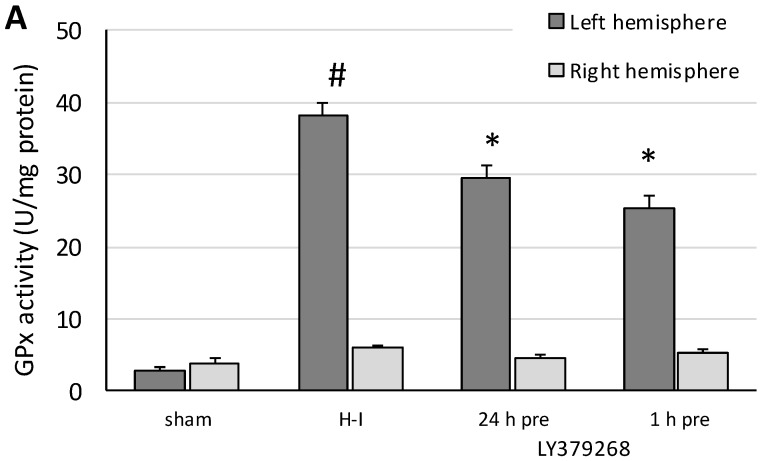
(**a**) The effect of LY379268 pretreatment on H-I-evoked changes in glutathione peroxidase activity and (**b)** the level of glutathione. The results are presented as the mean ± SD, *n* = 3–5; #—significantly different from the sham-operated group, *p* < 0.005; *—significantly different from the H-I group, *p* < 0.005.

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
