# Peer review of "Pretreatment with Group II Metabotropic Glutamate Receptor Agonist LY379268 Protects Neonatal Rat Brains from Oxidative Stress in an Experimental Model of Birth Asphyxia"

_brainsci, 2018, doi:10.3390/brainsci8030048_

Round 1

Reviewer 1 Report

Authors demonstrated that the mGluR II agonist (LY379268) able to reduce the infarct area in hemisphere by to reducing the levels of ROS, SOD, catalase, Glutathione peroxidase (GPx) that are known to increase during H-I condition. Same time the drug is able to restore the levels of GSH, that is known to reduce during H-I condition.

However, there are several problems persist in the study that dampens one’s overall enthusiasm.  

1)    The first issue is novelty in the overall manuscript.  As authors suggested in introduction (Line 73-76), the previous study has shown the neuroprotective effects of mGluR II agonist in asphyxia model and this happens through reduced levels of glutamate release. I don’t see novelty other than using agonist before H-I induction, which is gain not good idea keeping physiologically relevance in mind. Authors has to mention when exactly they apply the inhibitor; is it first LY injection followed by exposure of hypoxic condition??

The only novelty in manuscript is that the agonist protects the neonatal brain from oxidative stress in asphyxia model.

2)     Another major issue for me is the abstract: it is not easy to follow, what exactly authors wanted to summarize from their findings. Introduction should be clear and crisp. Example: it is not necessary keep this sentence ‘’The weight deficit of the ischemic brain hemisphere, reactive oxygen species (ROS) content, activity of antioxidant enzymes and the concentration of reduced glutathione (GSH) were measured’’

3)    This sentence is not clear to me at all; Line 25-28 ‘’The results suggest that the neuroprotective mechanisms triggered by activation of mGluR2/3 24 or 1 h before H-I act rather through the inhibition of oxidative stress and ROS production that results from the inhibition of glutamate release and decrease in its extracellular concentration’’. Also 24 or 1 hr can be removed from the sentence.

4)    Line 60-62:  Ischemia, hypoxia, oxidative stress or oxidative phosphorylation inhibition can activate ischemic tolerance when applied below the damage threshold [5-7]; however, such manipulations are potentially dangerous and, therefore, are not used in a clinical setting. This sentence is also not clear to me, please rewrite the sentence.

5)    Is it possible for authors to show western blot quantification for levels of oxidative stress markers (conditions Sh, H-I, LY+H-I)?

6)    It is important for Authors to show the efficiency of drug by measuring the activity of mGluR II, and dosage response. 

7) Figures for SOD, CAT, GPx levels can go into one figure rather than separate figures.

8) Title of manuscript could be even better.

Author Response

Comments and Suggestions for Authors

Authors demonstrated that the mGluR II agonist (LY379268) able to reduce the infarct area in hemisphere by to reducing the levels of ROS, SOD, catalase, Glutathione peroxidase (GPx) that are known to increase during H-I condition. Same time the drug is able to restore the levels of GSH, that is known to reduce during H-I condition.

We would like to thank the reviewer for a constructive review. The revised version of our manuscript address most of the reviewer’s  comments and suggestions.

However, there are several problems persist in the study that dampens one’s overall enthusiasm.  

The first issue is novelty in the overall manuscript.  As authors suggested in introduction (Line 73-76), the previous study has shown the neuroprotective effects of mGluR II agonist in asphyxia model and this happens through reduced levels of glutamate release.

 I don’t see novelty other than using agonist before H-I induction, which is gain not good idea keeping physiologically relevance in mind.

The only novelty in manuscript is that the agonist protects the neonatal brain from oxidative stress in asphyxia model.

We allow ourselves to disagree with the reviewer. Our manuscript presents for the first time the results of LY379268 pretreatment on H-I in an aspect of oxidative stress. There is a very small number of publications concerning the effect of mGlu2/3 agonists pretreatment on ischemia or H-I induced neurodegeneration and none of them discuss oxygen stress aspects. mGluR2/3 agonists are recently considered as drugs used in schizophrenia and anxiety treatment. Therefore, we think that our manuscript brings quite new and interesting data that may help in searching for new H-I  therapies and consider mGluR2/3 agonists as a potential medicines in H-I treatment.

Authors has to mention when exactly they apply the inhibitor; is it first LY injection followed by exposure of hypoxic condition??

The animals received a single injection at 24 h or 1 h before H-I. This is more precisely indicated in our revised manuscript.

2)     Another major issue for me is the abstract: it is not easy to follow, what exactly authors wanted to summarize from their findings. Introduction should be clear and crisp. Example: it is not necessary keep this sentence ‘’The weight deficit of the ischemic brain hemisphere, reactive oxygen species (ROS) content, activity of antioxidant enzymes and the concentration of reduced glutathione (GSH) were measured’’

The abstract has been changed according to the reviewer suggestions

3)    This sentence is not clear to me at all; Line 25-28 ‘’The results suggest that the neuroprotective mechanisms triggered by activation of mGluR2/3 24 or 1 h before H-I act rather through the inhibition of oxidative stress and ROS production that results from the inhibition of glutamate release and decrease in its extracellular concentration’’. Also 24 or 1 h can be removed from the sentence.

We changed this part of the text and we hope that now the information we wanted to bring to the notice in this sentence is clear.

4)    Line 60-62:  Ischemia, hypoxia, oxidative stress or oxidative phosphorylation inhibition can activate ischemic tolerance when applied below the damage threshold [5-7]; however, such manipulations are potentially dangerous and, therefore, are not used in a clinical setting. This sentence is also not clear to me, please rewrite the sentence.

This sentence was rewritten.

5)    Is it possible for authors to show western blot quantification for levels of oxidative stress markers (conditions Sh, H-I, LY+H-I)?

We thank the reviewer for this suggestion. In presented in this manuscript experiments we measured several of the main oxidative stress markers: ROS production, SOD activity and GSH level. All these measurements were made using commonly used biochemical methods. Unfortunately, in this set of experiments we did not measured the expression of antioxidant enzymes. As we were given a very short time for corrections, completion of suggested data was rather impossible.  We will try to complete the data and place them in one of our future manuscripts.

6)    It is important for Authors to show the efficiency of drug by measuring the activity of mGluR II, and dosage response. 

We thank the reviewer for this comment. We mentioned in the methods that the LY379268 concentration used in our experiments was chosen on the basis of previous publications by Cai et al. (1999) and Bond et al. (2000). The paper by Cai et al. (NeuroReport, 1999)  presents dose response data which served us as a reliable indication.

7) Figures for SOD, CAT, GPx levels can go into one figure rather than separate figures.

We thank the reviewer for this suggestion. However, with all the respect to the reviewer, we would like to leave these figures as they are. From our experience, the figures composed of several graphs are usually less readable.

8) Title of manuscript could be even better.

We thank the reviewer for this suggestion. We changed the manuscript title to more precise. We hope that the present title will meet the acceptance of the reviewers and the editor.

Reviewer 2 Report

Major Points:

1

The investigators conclude that the neuroprotection may be due to a suppression of ROS production. Could the investigators please expand on potential mechanisms for this decrease in ROS production? Is the ROS mitochondrial derived ROS?

2

Could the investigator please comment on whether post H-I treatment would be expected to result in protection.

3

While SOD, catalase and GPX activity decreased with LY379268 could the investigators please comment on whether other antioxidant enzymes could potentially be increased in response to LY379268.

Author Response

1

The investigators conclude that the neuroprotection may be due to a suppression of ROS production. Could the investigators please expand on potential mechanisms for this decrease in ROS production? Is the ROS mitochondrial derived ROS?

We thank the reviewer for this suggestion. In the present version of the manuscript we expanded the discussion including additional possibilities of LY379268 evoked decrease in ROS production after H-I.

DCF reaction measures all radical oxygen species present in the sample, which are not only ROS derived from mitochondria but also from lipids peroxidation and others.

2

Could the investigator please comment on whether post H-I treatment would be expected to result in protection.

We thank the reviewer for this comment. The effect of mGluR2/3 agonists applied after H-I was investigated by several scientists, mostly by Bond et al. and Cai et al. The results of their study show the neuroprotective effect mGluR2/3 agonists including LY379268. This information is included in the discussion of our results in this manuscript.

3

While SOD, catalase and GPX activity decreased with LY379268 could the investigators please comment on whether other antioxidant enzymes could potentially be increased in response to LY379268.

We thank the reviewer for this suggestion. In the new version of our manuscript we tried to include information considering other antioxidant enzymes and connect them with our results. However, the additional experiments to explain the involvement of these enzymes in the observed LY379268 effect are necessary.

Round 2

Reviewer 1 Report

Still I have problem with the article.

From abstract: ‘The results suggest that the neuroprotective mechanisms triggered by activation of mGluR2/3 before H-I act through the inhibition of ROS production and reduction of oxidative stress, which results from the decrease of glutamate release and its extracellular concentration’’

Novelty of the study: agonist is able to reduce not only glutamate release but also oxidative stress and protect brain. 

My major concern is, how authors can say that the mGluR agonist protects the brain by decreasing glutamate release followed by decrease in oxidative stress??  or vice versa..

Until unless authors provided solid evidence for existence of glutamate toxicity and oxidative stress in their H-I conditions and usage of drug ameliorates both of these. Please see the reference Proc Natl Acad Sci U S A. 2004 May 18;101(20). This is the basic criteria for experimental set up.

           It is also not clear what is the mode mechanism of action (agonist) for decreased oxidative stress?

Author Response

Response to the reviewer:

From abstract: ‘The results suggest that the neuroprotective mechanisms triggered by activation of mGluR2/3 before H-I act through the inhibition of ROS production and reduction of oxidative stress, which results from the decrease of glutamate release and its extracellular concentration’’

 Novelty of the study: agonist is able to reduce not only glutamate release but also oxidative stress and protect brain. 

My major concern is, how authors can say that the mGluR agonist protects the brain by decreasing glutamate release followed by decrease in oxidative stress??  or vice versa..

In the discussion we have presented the molecular pathway leading from excessive glutamate release to ROS formation and oxidative stress development. Basing on our results indicating that LY379268 reduces  ROS production after H-I we concluded that there is a direct connection between basic mGluR2/3 function (decrease in Glu release) and observed effects. We do not exclude that other mechanisms take part in the neuroprotective effect of LY379268  but these mechanisms were not the subject of our investigation.

Until unless authors provided solid evidence for existence of glutamate toxicity and oxidative stress in their H-I conditions and usage of drug ameliorates both of these. Please see the reference Proc Natl Acad Sci U S A. 2004 May 18;101(20). This is the basic criteria for experimental set up.

We thank the reviewer for this comment. We found two interesting papers in the recommended issue. We presume that the article concerning the role of mGluR in excitotoxicity and oxidative stress is the one. The mentioned reference is indeed interesting and may be an example of  a well done scientific job. However, the experiments presented in this paper were performed on oligodendrocyte culture and the authors concentrated mostly on group I mGluRs.

The experimental model used in our experiments is well characterized and served to many researches. There is no doubt that this model of H-I results in glutamate toxicity and oxidative stress (for review see Vannucci and Hagberg, J Exp Biol, 2004;207(Pt 18):3149-54). The publication by Makarewicz et al. (Folia Neuropathol. 2014, 52(3): 270-284) derived from our laboratory and performed on the same H-I model also indicates on involvement of glutamate excitotoxicity and oxidative stress. We also showed the presence of oxidative stress and changes in ROS level as well as in antioxidant enzymes activity after H-I in two other papers. With all respect, we think that our previous and present results give the evidence for glutamate toxicity and oxidative stress in used H-I model and also showed the influence of used drugs on these factors.

           It is also not clear what is the mode mechanism of action (agonist) for decreased oxidative stress?

We agree with the reviewer that the exact mechanism still needs explanation. Therefore in the discussion we suggested that the close investigation of the effect of LY379268 on the early events leading to oxidative stress, like NO accumulation and the engagement of thioredixin, needs to be done.